# Plasmonic high-entropy carbides

**Arrigo Calzolari** [1] ✉, **Corey Oses** [2,3], **Cormac Toher**[3,4], **Marco Esters** [2,3], **Xiomara Campilongo** [3], **Sergei P. Stepanoff**[5], **Douglas E. Wolfe**[5] & **Stefano Curtarolo** [2,3] ✉

Discovering multifunctional materials with tunable plasmonic properties, capable of surviving harsh environments is critical for advanced optical and telecommunication applications. We chose high-entropy transition-metal carbides because of their exceptional thermal, chemical stability, and mechanical properties. By integrating computational thermodynamic disorder modeling and time-dependent density functional theory characterization, we discovered a crossover energy in the infrared and visible range, corresponding to a metal-to-dielectric transition, exploitable for plasmonics. It was also found that the optical response of high-entropy carbides can be largely tuned from the near-IR to visible when changing the transition metal components and their concentration. By monitoring the electronic structures, we suggest rules for optimizing optical properties and designing tailored high-entropy ceramics. Experiments performed on the archetype carbide $HfTa_4C_5$ yielded plasmonic properties from room temperature to 1500K. Here we propose plasmonic transition-metal high-entropy carbides as a class of multifunctional materials. Their combination of plasmonic activity, high-hardness, and extraordinary thermal stability will result in yet unexplored applications.

Manufacturability, the need to design products that are easy to manufacture, is one of the highest priorities in materials science. Yet, it remains a challenge in industrial production[1]. Additionally, high costs and sourcing raw materials further complicates the issue. One possible solution lies in multifunctional compounds, combining in a single system features usually attained in different classes of materials and fabrication processes. In particular, the discovery of optical-mechanical multifunctional materials would enable extending optical applications (e.g., radar and satellite, thermo-optical generators, thermophotovoltaics, magnetic recording) that require optical plasmon-based solutions (e.g., emitters[2], modulators and detectors[3,4], switches[5], broadband absorbers[6,7], etc.) in harsh environments (e.g., high temperature and pressure[8,9], and chemical or mechanical abrasion[10]).

Plasmons—electronic collective excitations controlled by external electric fields—are exploited: i. to confine the electromagnetic radiation to nanometric regions shorter than the diffraction limits[11,12], ii. to

amplify local electric fields[13], and iii. to generate "extraordinary" propagating waves in metamaterials and hyperbolic media[14,15]. Because of these capabilities, plasmonic materials[16] are growing their appeal in a wide range of applications including light harvesting[17], biosensing[18], neuroscience[19], telecommunications[20], hyperbolic metamaterials[21,22], and quantum optics. Other classes of thermal devices, e.g., photo-thermal emitters[23,24] and heat-assisted magnetic recording (HAMR)[25], exploit the de-excitation of plasmons and the consequent localized high-temperature gradient.

No single material satisfies the requirements of all the possible applications[26]. Standard plasmonic metals (Au, Ag, Cu) are ductile, have a reduced thermal stability, are not CMOS (Complementary Metal-Oxide Semiconductor) compatible, and are difficult to pattern into thin films or layers, making them unsuitable for high temperature or mechanical stress-working conditions[27]. Doped metal-oxides (AZO, ITO)[28–30] and transition-metal nitrides (TiN and ZrN)[31–34] have been proposed as plasmonic materials in the visible range. The former have

[1]CNR-NANO Research Center S3, Modena, Italy. [2]Department of Mechanical Engineering and Materials Science, Duke University, Durham, NC, USA. [3]Center for Autonomous Materials Design, Duke University, Durham, NC, USA. [4]Department of Materials Science and Engineering and Department of Chemistry and Biochemistry, University of Texas at Dallas, Richardson, TX, USA. [5]Applied Research Laboratory, The Pennsylvania State University, University Park, PA, USA. ✉e-mail: arrigo.calzolari@nano.cnr.it; stefano@duke.edu

tunable plasmonic energies but poor mechanical resistance; while the latter have good mechanical properties, but non-tunable plasmonic energy. Transition metal (TM) carbides, such as TiC, TaC, and WC, are simple systems, easy to grow, CMOS compatible, with high hardness, high thermochemical stability, and have been largely used for coating and anti-corrosion applications. Yet, they have poor optical properties (e.g., reflectance, a summary of the optical and electronic properties of TM-carbides is reported in Supplementary Figs. 1, 2).

Among TM-carbides, only TaC exhibits a plasmonic excitation in the visible range, with all others being non-plasmonic[35]. This carbide has a plasmonic resonance at $E = 2.7$ eV, high Vickers hardness ($H_V = 27.6$ GPa), and represents an example of a multifunctional plasmonic-mechanical material. However, in view of the large application range, the need for tunability of optical properties becomes crucial[36]. In general, the possibility to combine tunable plasmonic properties with thermal stability and/or mechanical resistance would enable high temperature devices, novel optically active refractory, coating or wearing materials for aerospace or satellite applications. While a few examples to combine optical properties and extreme hardness have been achieved in superlattice metamaterials[37,38], ours is an attempt to combine all the useful properties in a multifunctional material.

Here, we propose plasmonic transition-metal high-entropy carbides (PHECs) as a class of multifunctional materials able to couple thermal stability, mechanical resistance, and plasmonic properties in the near-IR and visible range. Indeed, they gather the good plasmonic properties of TaC, the superior mechanical properties of TM-carbides, and the tunability given by compositional freedom[39]. HECs were discovered in 2018 by some of the authors[40], through an entropy-forming ability (EFA) descriptor, calculated on a set of randomized structures[41]. Along with high thermodynamic stability[39,40,42,43], HECs showed increased hardness[40,42,44–46], strength[42,47,48], toughness[42,48] and resistance to creep[49]—attributes that are maintained to high temperatures[42,47–49]—along with improved wear resistance[39,50] and better resistance to oxidation and corrosion[39,51–54]. This work extends the discovery algorithm beyond synthesizability and hardness by employing time-dependent density-functional theory to calculate the dielectric function. The multi-element character of the high-entropy carbides offers a natural testbed to investigate the role of chemical composition and stoichiometry on the tunability of the plasmon excitation and of the plasma energy.

## Results
### Materials response
The response of materials to incoming electromagnetic waves is described by the complex dielectric function $\hat{\epsilon}(\mathbf{q}, E) \equiv (\epsilon_r + i\epsilon_i)$, where $\mathbf{q}$ is the transferred momentum and $E$ is the energy of the incoming field[55]. The real part, $\epsilon_r$, relates to polarization effects and

distinguishes between dielectrics ($\epsilon_r > 0$) and metals ($\epsilon_r < 0$). The imaginary part, $\epsilon_i$, is associated with dissipation of energy into the medium. For bulk systems, the electron-energy loss spectra (EELS) is defined as $-\text{Im}[\epsilon_j^{-1}(\mathbf{q}, E)]$ and it describes the energy loss due to inelastic electron scattering upon electromagnetic irradiation. In the low-loss energy region ($E < 50$ eV), the spectrum of EELS provides information about electronic structure, optical, and plasmonic properties. The plasmon excitation energies are the poles of $\hat{\epsilon}(\mathbf{q}, E)$ and correspond to EELS's peaks. Experimentally, plasmons are determined by measuring the characteristic electron-energy loss. The evaluation of $\hat{\epsilon}(\mathbf{q}, E)$ and the corresponding EELS completely characterizes the plasmonic features, and provides a direct pathway to comparison with experimental results[32].

Here, we adopt a high-throughput numerical approach that couples thermodynamic methods[56], for the evaluation of the atomic structure and stability of PHECs, with first principles approaches based on time-dependent density-functional perturbation theory for the characterization of optical and plasmonic properties. In view of the optical properties of native TaC, we consider, as an example, the case of $HfTa_4C_5$, which is a well-established compound with exceptional thermal and mechanical properties.

### Plasmonics and disorder modeling
The macroscopic disorder is factorized as the sum of many microscopic configurations, also called *tiles*, which are determined by the Partial Occupation module (POCC) method[41], as implemented within the AFLOW computational materials design framework[57]. The method generates a factor-group weighted Boltzmann ensemble of Hermite normal form superlattices, having the minimum size compatible with the required stoichiometry and accuracy. Non-unique configurations are removed through symmetry[58] and identification[59] considerations. The capability of this approach in reproducing the electron and phonon density of states has already been proved[60]. Here we extend it to optical spectra of disordered systems.

For each optimized POCC *j*-structure, we calculate $\hat{\epsilon}_j(\mathbf{q}, E) \equiv (\epsilon_r + i\epsilon_i)_j$, and the corresponding EELS, $-\text{Im}[\epsilon_j^{-1}(\mathbf{q}, E)]$. Unless otherwise specified, we consider only the optical limit at $\mathbf{q} = 0$. Then, the spectra of the disordered material are obtained as an ensemble average on the POCC *j*-structures (see Methods Section). The poles (peaks) of the averaged dielectric function (EELS) defines the spectral position of the plasmon resonance. As an accuracy testbed, we compare our results to an archetype of plasmonic materials, the disordered solid-solution AuAg and the fcc constituents Au and Ag. The results are summarized in Supplementary Fig. 3. The excellent agreement between simulated and experimental findings confirms the accuracy of the POCC method in calculating optical properties of disordered systems. This is not surprising: POCC is expected to well

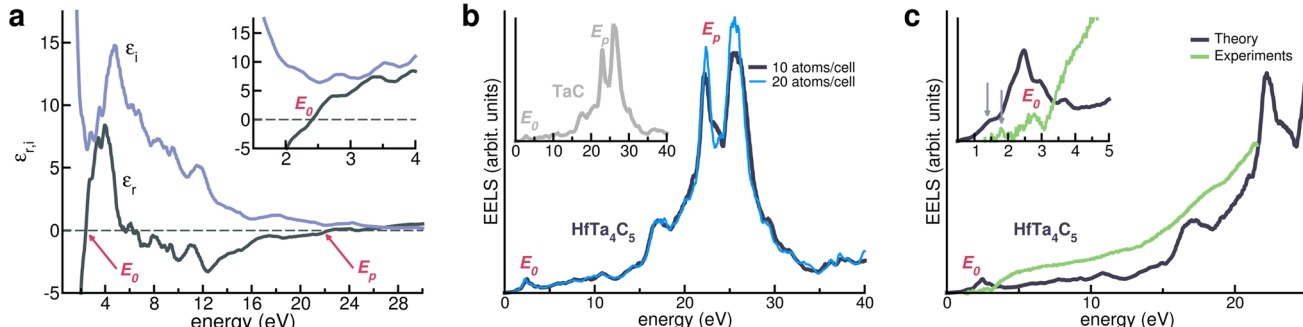

**Fig. 1 | Optical properties of plasmonic HfTa₄C₅. a** Real ($\epsilon_r$, dark gray) and imaginary ($\epsilon_i$, light purple) part of the complex dielectric function. $E_0$ and $E_p$ indicate the crossover energy and the plasmon energy, respectively. Inset zooms on low-energy range of the spectrum. **b** Simulated electron-energy loss spectrum (EELS) calculated by assuming two sets of structures with 10 (black) or 20 (cyan) atoms per POCC cell at $T = 0$ K. Simulated EELS of TaC crystal (light gray) is included in inset for comparison. **c** Comparison between simulated (dark gray) and experimental (green) EELS spectra. Inset zooms on low-energy range of the spectrum. Source data are provided as a Source Data file.

reproduce all disorder properties which have characteristic lengths larger than the one of the radial distribution function, which is captured by the size cutoff of the POCC cells. On the basis of these results on well-known plasmonic materials, we use this method to predict the plasmonic properties of high-entropy carbides.

The complex dielectric function of $HfTa_4C_5$ is shown in Fig. 1a. At low energy ($E \rightarrow 0$) the real part of the dielectric function $\epsilon_r$ is negative, while the imaginary part $\epsilon_i$ is positive and diverging, as in free-electron (i.e., Drude-like) metals. Overall, $\hat{\epsilon}(E)$ maintains a metal-like behavior in

the energy range from $E \sim 12.0$ eV to $E_p = 22.36$ eV. $E_p$ is the plasma energy of the system and corresponds to the excitation of a *bulk plasmon*, i.e., to a collective oscillation of the total free charge density. $\epsilon_r$ becomes definitively positive (i.e., dielectric behavior) for $E > E_p$. Despite interesting quasiparticle properties, bulk plasmons have excitation energies which are too high for any realistic application. On the contrary, optical properties in near-IR/visible part of the spectrum are more useful. In particular, at $E_0 = 2.41$ eV and upon the application of an external electromagnetic field, the system undergoes an abrupt change of the optical response from metal- to dielectric-like. $E_0$–so-called *crossover energy*–identifies the energy at which the real part of the dielectric function switches sign ($- \rightarrow +$, see Fig. 1). This behavior results from a balance between intraband transitions, responsible for the negative Drude tail and associated to free-electron density, and the excitation of interband transitions giving positive contributions to $\epsilon_r$, which turns positive at $E_0$. In the case of $HfTa_4C_5$ at the crossover energy, $\epsilon_i(E_0)$ also has a minimum (see inset in Fig. 1a). This corresponds to the possibility of exciting a collective electronic oscillation of a reduced (*screened*) part of the free-electron density, known as a *screened plasmon* resonance. In this case, $E_0$ corresponds also to the low-energy plasma energy. The condition $\hat{\epsilon}(E) \simeq 0$ results in a peak in the electron-energy loss spectra.

The simulated optical EELS (Fig. 1b) is dominated by the high-energy peak at $E_p$ and by the low-energy peak at $E_0$, which are the *fingerprints* of the bulk and screened plasmons, respectively. We conclude that $HfTa_4C_5$ combines well a plasmonic response in the visible range with superior thermal and mechanical properties, making it the first instance of a multifunctional disordered carbide. The size and the number of the adopted POCC structures do not affect this result. The simulation of disordered $HfTa_4C_5$ with a double number of atoms per cell (cyan line in Fig. 1b) reproduces all the spectral features of the original system (black line) with, e.g., a difference in the crossover energy $E_0$ smaller than 0.15 eV.

To corroborate our claims, a set of experimental EELS measurements on the archetype high-temperature carbide $HfTa_4C_5$ was performed from room temperature to 1500 K (Methods and

**Table 1 | List of simulated TM-carbides and PHEC structures at varying composition**

| # | Carbide | $E_0$ | $h(E_0)$ | $E_p$ | $g_4$ $g_5$ $g_6$ | Class |
|---|---------|-------|----------|-------|----------|-------|
|   | TaC | 2.72 | 0.113 | 23.07 | 0 5 0 | |
|   | $HfTa_4C_5$ | 2.41 | 0.112 | 22.36 | 1 4 0 | |
| 1 | $HfNbTiVZrC_5$ | 1.21 | 0.012 | 21.68 | 3 2 0 | $3g_4$ |
| 2 | $HfTaTiVZrC_5$ | 1.29 | 0.029 | 21.74 | 3 2 0 | $3g_4$ |
| 3 | $HfNbTaTiZrC_5$ | 1.42 | 0.071 | 21.63 | 3 2 0 | $3g_4$ |
| 4 | $HfTaTiWZrC_5$ | 1.77 | 0.042 | 21.91 | 3 1 1 | $3g_4$ |
| 5 | $NbTaTiVZrC_5$ | 1.52 | 0.035 | 22.20 | 2 3 0 | $3g_5$ |
| 6 | $HfNbTaTiVC_5$ | 1.59 | 0.039 | 22.50 | 2 3 0 | $3g_5$ |
| 7 | $HfNbTaVZrC_5$ | 1.62 | 0.038 | 21.84 | 2 3 0 | $3g_5$ |
| 8 | $NbTaTiVWC_5$ | 1.96 | 0.045 | 23.43 | 1 3 1 | $3g_5$ |
| 9 | $MoNbTaVWC_5$ | 2.35 | 0.022 | 24.35 | 0 3 2 | $3g_5$ |
| 10 | $CrMoTiVWC_5$ | 1.77 | 0.016 | 24.68 | 1 1 3 | $3g_6$ |
| 11 | $CrMoNbVWC_5$ | 2.08 | 0.012 | 24.51 | 0 2 3 | $3g_6$ |
| 12 | $CrMoTaVWC_5$ | 2.21 | 0.010 | 24.58 | 0 2 3 | $3g_6$ |
| 13 | $CrMoNbTaWC_5$ | 2.48 | 0.012 | 24.25 | 0 2 3 | $3g_6$ |
| 14 | $HfNbTaTiWC_5$ | 1.96 | 0.060 | 22.55 | 2 2 1 | $2g_4 2g_5$ |

The crossover ($E_0$) and the bulk plasmon energies ($E_p$) are expressed in eV. The effective EELS intensity coefficient ($h$) is calculated at $E = E_0$ and expressed in arbitrary units. PHECs are ordered in four Classes depending on the the number of composing TMs from group 4 $g_4$ (Ti, Nb, Hf), group 5 $g_5$ (V, Nb, Ta), and group 6 $g_6$ (Cr, Mo, W) of the periodic table.

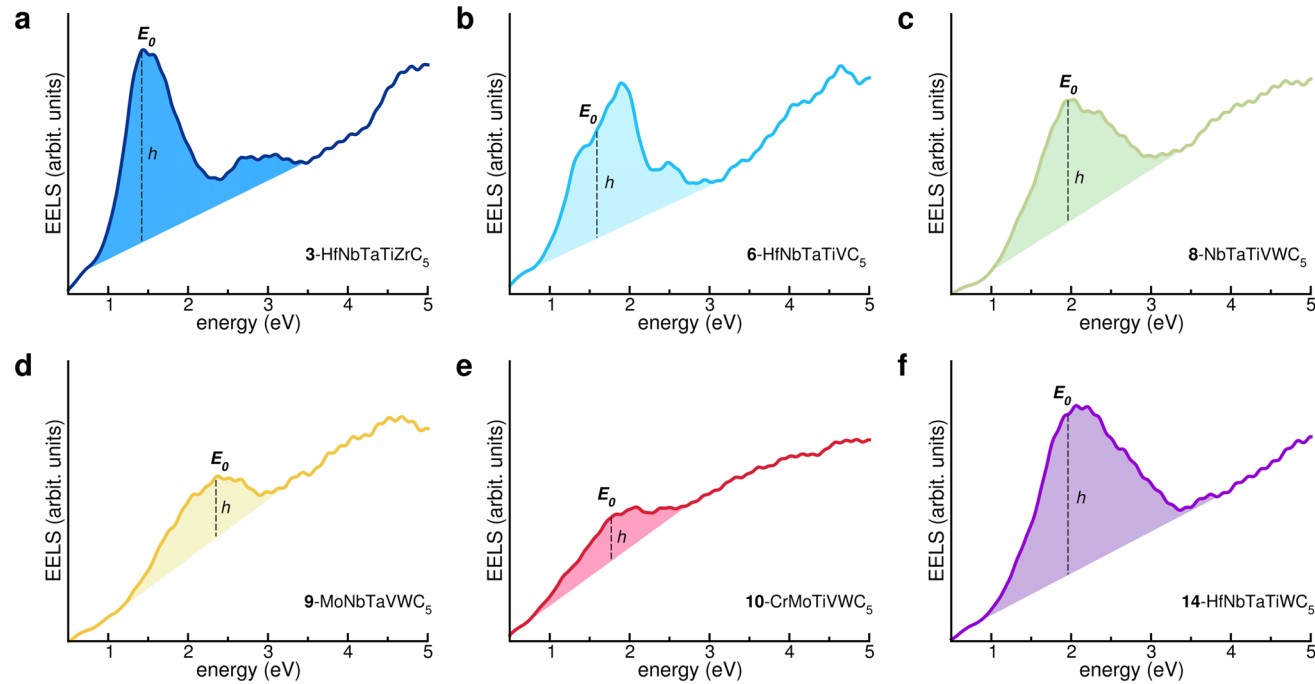

**Fig. 2 | Simulated EELS spectra for selected HECs.** $E_0$ and $h$ indicate the crossover energy and the effective intensity of the EELS spectra corresponding to energy $E = E_0$. **a** 3-$HfNbTaTiZrC_5$. **b** 6-$HfNbTaTiVC_5$. **c** 8-$NbTaTiVWC_5$. **d** 9-$MoNbTaVWC_5$. **e** 10-$CrMoTiVWC_5$. **f** 14-$HfNbTaTiWC_5$. Source data are provided as a Source Data file.

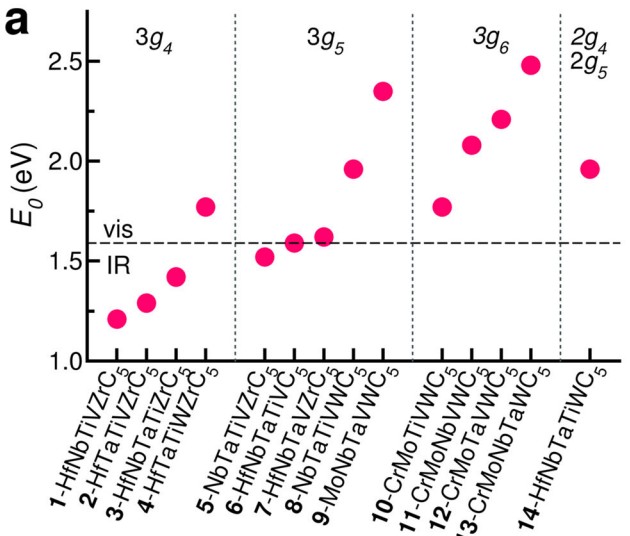

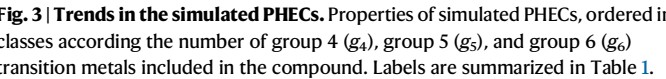

**Fig. 3 | Trends in the simulated PHECs.** Properties of simulated PHECs, ordered in classes according the number of group 4 ($g_4$), group 5 ($g_5$), and group 6 ($g_6$) transition metals included in the compound. Labels are summarized in Table 1.

**a** Crossover energy ($E_0$), with horizontal dashed line indicating the IR/visible range. **b** effective EELS intensity $h$. Source data are provided as a Source Data file.

Supplementary Information). Figure 1c compares the simulated (dark gray) and the experimental (green) loss functions. The two curves concur (except for minor details), reproducing the main optical features of the material, in particular the screened plasmon with its maximum at $E_0 = 2.7$ eV−very close to the simulated value. In addition, the experimental spectrum has similar features to plasmonic TiN, characterized by the same EELS techniques[61]. The theoretical spectra also reproduces the low-intensity shoulder (inset, vertical arrow) close to the main $E_0$ peak.

The comparison between HfTa$_4$C$_5$ and crystalline TaC (inset, panel b) indicates that the loss functions of the two systems are very similar, except for the values of $E_0$ and $E_p$ (Table 1). The inclusion of Hf imparts a red-shift of both the main spectroscopic features, suggesting a major role of composition and stoichiometry in the optical properties of the system. To exploit the composition-driven-shift, we try to understand its cause so we can then search for PHECs with (1) plasmonic resonances in the IR-vis range and (2) tunable crossover energy.

### Understanding composition-driven red-shift

Let us consider two sets of virtual-crystal compositional studies, focused on the separate effects of stoichiometry and chemical composition. *Stoichiometry:* we consider Hf$_x$Ta$_{5-x}$C$_5$ with $x = [0−5]$, where $x = 0$ and $x = 5$ correspond to pure TaC and HfC, respectively. *Chemical composition:* we choose different transition metals in $M$Ta$_4$C$_5$, where $M \in \{$Ti, Zr, Hf, V, Nb, Cr, Mo, W$\}$.

The tests have been calculated starting from a single reference POCC structure of HfTa$_4$C$_5$ and substituting Ta atoms with different transition metals. The results, summarized in Supplementary Figs. 3 and 4, indicate that the increase of Hf content is followed by degradation of the screened plasmon features up to the limiting case of HfC, whose optical properties are included in the Supplementary Information (Supplementary Fig. 5) for comparison. The latter does not exhibit any low-energy peak in the loss function, similar to the case of HfC rocksalt structure (Supplementary Fig. 1).

Substitution of Hf with other transition metals can have different effects on the optical spectrum: i. energy shifts for $E_0$, i.e., red- and blue-shifts for group 4 and 6 metals, respectively; ii. plasmonic resonance deterioration with 3d metals (especially V and Cr), with strongly reduced intensity of the EELS peak; and iii. broadening of the EELS peak with group 6 metals, indicating that Ta, Zr, Hf, Nb, and W seem to be the most promising elements to

obtain plasmonic carbide materials; while V, Cr, and Mo can be neglected.

### Plasmonic high-entropy carbides

To corroborate the finding, we consider a set of 14 PHECs, whose stability and crystalline phase was recently demonstrated[40,44]. These are six element alloys: carbon and a combination of five different transition metals among the nine elements of groups 4 (Ti, Zr, Hf), 5 (V, Nb, Ta), and 6 (Cr, Mo, W) of the periodic table. The complete list of simulated compounds is reported in Table 1, where an integer label ($\{\mathbf{1}, \cdots, \mathbf{14}\}$) is also assigned to each system for simplicity. All EELS plots exhibit spectroscopic characteristics similar to the HfTa$_4$C$_5$ case of Fig. 1, e.g., the characteristic double peak in the loss function corresponding to the bulk plasmon resonance $E_p$ in the far UV, and a crossover energy $E_0$ in the IR-visible part of the spectrum (see Table 1 for numerical values and Fig. 2 for loss spectra).

Let us now focus on the low-energy part of the spectrum ($E < 5.0$ eV) in which we refer to *screened plasmons* simply as plasmons. Even though a spectroscopic resonance around $E_0$ is evident in all systems, the energy position, the width and the intensity of the peaks vary with composition. Peaks in systems **3**-HfNbTaTiZrC$_5$ and **14**-HfNbTaTiWC$_5$ are high, sharp and well defined. They reduce to low intensity and broad shoulders in systems **9**-MoNbTaVWC$_5$ and **10**-CrMoTiVWC$_5$. In systems **6**-HfNbTaTiVC$_5$ and **8**-NbTaTiVWC$_5$, peaks have intermediate shape.

Figure 3a shows the energy variation of $E_0$ as a function of composition. The systems are grouped into four classes with respect to the TM groups (Table 1). The first class, $3g_4$ in Fig. 3, maximizes the number of group 4 elements (Ti, Zr, Hf); the second, $3g_5$, maximizes the group 5 metals (V, Nb, Ta); the third, $3g_6$, those of group 6 (Cr, Mo, W), with the remaining two elements per system from other groups; the last ($2g_42g_5$) has a mixed composition, with two elements from group 4 and 2 from group 5. Figure 3a shows some evident trends. *Wide ranges.* $E_0$ spans a remarkable range of energies from near-IR to visible. This is quite interesting, as the possibility to control and tune the optical properties of materials by varying the composition is of critical relevance in the development of plasmonics and nanophotonics applications. *Intragroup increase.* Within each group, $E_0$ increases monotonically with the increase of the atomic number $Z$ of the constituents. For example, systems **1**–**3** have in common three elements of group 4 (Ti, Zr, Hf) and differ for the remaining two elements from the

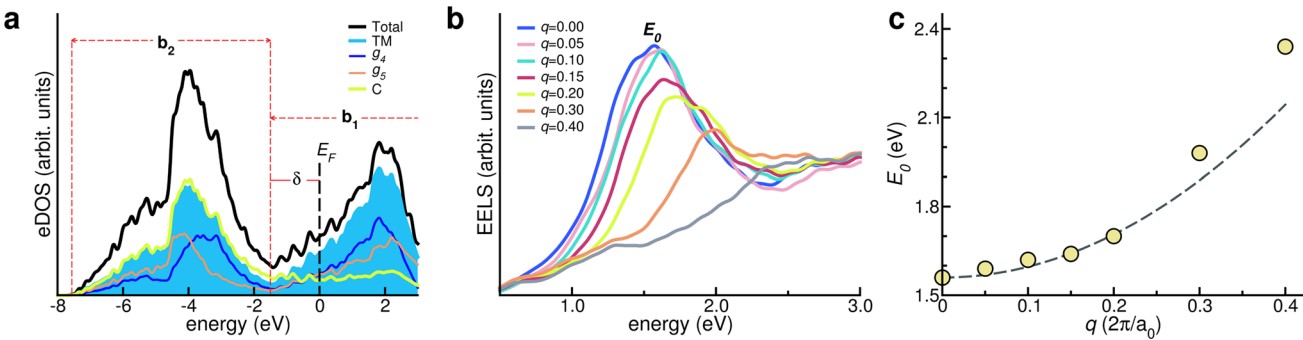

**Fig. 4 | Optical properties of plasmonic 3-HfNbTaTiZrC₅ system. a** Total (black area), TM- (cyan area), $g_4$ - (thin blue line), $g_5$ - (thin orange line) and C-projected (thick green line) eDOS. Zero energy reference is set to the Fermi level ($E_F$). $\mathbf{b_1}$, $\mathbf{b_2}$ and $\delta$ indicate the band manifolds and the energy difference discussed in the text. **b** Simulated EELS for a representative POCC structure at increased transferred momentum $q$ in units of $2\pi/a_0$. **c** Plasmon energy $E_0(q)$ dispersion relation. Yellow circles are the calculated crossover energies from **b**) while the black dashed line is the corresponding quadratic fit. Source data are provided as a Source Data file.

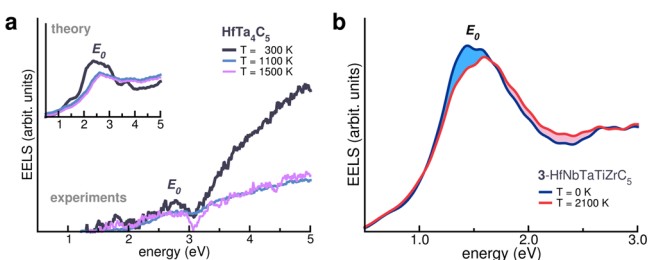

**Fig. 5 | Thermal evolution of plasmonic properties. a** Experimental EELS spectra of HfTa₄C₅ as a function of temperature. Inset reports the corresponding theoretical spectra, evaluated at the same temperatures. **b** Simulated EELS spectra of 3-HfNbTaTiZrC₅ at different temperatures. Source data are provided as a Source Data file.

group 5, V-Nb (**1**), V-Ta (**2**), and Nb-Ta (**3**). Indeed, $E_0$ increases from system-**1** to **3** along with the atomic number of the constituents. *Intergroup increase.* Across different groups, the higher the $Z$ of the elements, the higher becomes the crossover energy. $E_0$ tends to increase moving from 3$g_4$ to 3$g_6$, i.e., increasing number of electrons. For mixed compositions, e.g., system **14**-HfNbTaTiWC₅, $E_0$ spans the intermediate energy range.

The heights and the widths of the $E_0$ peaks are also indicative of the properties of the plasmonic response: sharp and intense peaks (small $\epsilon_i$) are representative of low-loss collective oscillations, i.e., long-living plasmons; broad and small peak shoulders (large $\epsilon_i$) indicate single-particle dissipation, i.e., no plasmon excitations. For the ideal plasmonic resonance, the entire complex dielectric function should be zero at $E_0$, like in the ideal free-electron metals, corresponding to a loss function being zero everywhere, except for $E_0$, where a sharp peak rises. Here, the non-zero value of the EELS corresponds to imaginary part of the dielectric function $\epsilon_i$ different than zero, indicating dissipative interband optical absorption. If the latter becomes predominant, $\epsilon_i$ does not show any minimum and the corresponding peaks in the EELS spectra cannot be associated to collective plasmon-like oscillations, but rather to single-particle optical transitions and energy loss dissipative processes. Here, $E_0$ would simply represent the threshold between metallic ($E < E_0$) and dielectric ($E > E_0$) behavior.

In order to estimate the intensity of the resonance peak with respect to background, we define an effective EELS intensity, $h$ in Fig. 2, calculated as the convex distance of the peak to the linearized spectra around $E_0$. This simple and phenomenological quantity allows us to separate plasmonic from the dissipative contributions (colored versus white areas in Fig. 2). The effective intensities are shown in Fig. 3b.

Systems **3**-HfNbTaTiZrC₅ and **14**-HfNbTaTiWC₅ have the highest $h$ and the lowest energy loss dissipation, i.e., good plasmonic properties. Systems **4**–**8** have mixed characters, with broad but still evident excitation peaks that can be associated to lossy plasmons. For the remaining systems, dissipation is too high and no collective oscillations can be associated to the $E_0$ resonances. From the analysis, we discover five important facts: i. the presence of Ta is necessary but not sufficient for obtaining good plasmonic PHECs; ii. the best results are obtained combining high-$Z$ elements from groups 4 and 5 (Zr, Hf, Nb, and Ta); iii. V, Cr and Mo are detrimental; iv. accurate choice of TMs allows for a modulation of the crossover energy over a large part of the spectrum; and v. the quasi-neutral effect of Ti and W on the final optical properties is also advantageous because it can be exploited to stabilize the PHEC and to change its mechanical or thermal properties.

The compositional variability of the plasmonic properties of PHEC can be interpreted in terms of their electronic structure. Figure 4a shows the electronic density of states, eDOS, of **3**-HfNbTaTiZrC₅, used as the testbed case. Near the Fermi energy, $E_F$, the eDOS has two main peaks: $\mathbf{b_1}$ has a predominant TM character (cyan area), it crosses the Fermi level and is responsible for the Drude character of the dielectric function; $\mathbf{b_2}$ has mixed character with contributions from both *sp* states of carbon (light green line) and the *d* states of TMs; and it is mainly responsible for the optical excitations from valence to conduction band.

The difference $\delta$ between the $\mathbf{b_2}$'s edge and the Fermi level can give an estimate of the crossover energy (Fig. 4a). Only $\mathbf{b_1}$ intraband transitions are possible for incoming radiation having energy $E < \delta$, while for $E > \delta$, both interband $\mathbf{b_2} \rightarrow \mathbf{b_1}$ and intraband $\mathbf{b_1} \rightarrow \mathbf{b_1}$ compete, leading to an increase of the real part of the dielectric function (dielectric screening) that crosses zero at $E_0$.

While this behavior is common to all simulated PHECs, details of the $E_0$ and $h$ are composition-dependent and can be related to the electronic structure of the parent binary TM-carbides (e.g., TiC, VC, NbC, WC, etc.), as shown in Supplementary Fig. 2[38]. The $\mathbf{b_1}$ and $\mathbf{b_2}$ peaks are observed in all the TM-carbides. Moving from group 4 to group 6 carbides, the higher number of electrons shifts the Fermi level deeper in the $\mathbf{b_1}$ band increasing distance $\delta$ from $\mathbf{b_2}$. This justifies the blue-shift of $E_0$ observed in Fig. 3a. In addition, moving from 3*d* to 5*d* carbides, more bands belonging to $\mathbf{b_1}$ cross $E_F$. This increases the relative amount of free-electron density and thus the intraband component of the excitation. This finding corroborates the enhanced plasmonic behavior in compounds with higher atomic number. In the specific case of Fig. 4a, the projection of the TM component of the eDOS on group 4 (blue line) and group 5 (orange line) elements highlights the shift of group 5 toward higher binding energies and the corresponding displacement of the $\mathbf{b_2}$ band from the Fermi level. This imparts an

increase of $\delta$ and the blue-shift of the crossover energy. In addition, the high metal-derived component of the eDOS at the Fermi level is responsible for the good plasmonic character of the excitation at $E = E_0$.

In the free-electron approximation, a plasmon-like excitation follows a parabolic dispersion of $E_0$, function of the transferred momentum $q$. Figure 4b shows the evolution of the crossover peak for different transferred momenta, for the testbed system 3-HfNbTa-TiZrC$_5$. Increasing $q$ causes a blue-shift of the $E_0$ energy and a flattening of the EELS, which becomes broader and less intense. The dispersion of the plasma energy $E_0(q)$ is almost quadratic (parabolic fit—dashed line—superimposed in Fig. 4c). This behavior is a fingerprint of the collective plasmon-like character of $E_0$. The upward energy dispersion reflects the broadening and the reduction of the intensity of the peak shown in panel b, and it is an indication of the deviation from the purely free-electron character of the system due to dissipative optical transitions.

Since HfTa$_4$C$_5$ and HECs have exceptional mechanical properties, high hardness, and super-high thermal stability, we investigate the thermal evolution of the plasmonic properties. Results for HfTa$_4$C$_5$ and system 3-HfNbTaTiZrC$_5$ are summarized in Fig. 5. Panel a shows the experimental and theoretical EELS of spectra HfTa$_4$C$_5$ at different temperatures, in the range T ∈ [300–1500] K. Increasing temperature produces minor changes to the main plasmonic properties: the spectral feature corresponding to the low-energy plasmon remains clearly recognizable up to T = 1500 K. The plasmon resonance is surprisingly stable even at high temperature, much higher than the standard plasmonic metals' melting points (e.g., Ag and Au). Besides, the increased temperature causes an expected slight reduction of the intensity and a small broadening of the $E_0$ peak, which may be attributed to an increase of interband effects. Simulations (inset, panel a) concur with experiments, representing plasmonic resonance even at high temperatures. It is worth noting that here the temperature is the *conformational temperature* used to average the POCC ensemble[41]. Beyond 1500K, it might be arduous to characterize experimental EELS due to equipment limitation. Experimental-computational agreement for HfTa$_4$C$_5$ makes us confident of the existence of plasmonic properties of 3-HfNbTaTiZrC$_5$ even at ultra-high temperature (panel b)—temperature should have a minor effect on the spectral feature, causing a small reduction of the maximum intensity of $E_0$, along with a broadening of the peak due to larger scattering effects. Since plasmons are purely electronic excitations, ionic temperature does not affect the energy position of the plasmonic band (i.e., the crossover energy), except for effects related to thermal expansion of the atomic structure[55] (see the Supplementary Information for further details). The results of Fig. 5 indicate the possibility to excite a plasmon resonance even at ultra-high-temperature, where structural resistance of HECs can be advantageous for applications. Temperature could still affect lifetime and de-excitation of the plasmons through dissipative electron-phonon scattering. Our approach does not include time evolution—de-excitation and related energy release—which is beyond the purpose of this work.

## Discussion

This article shows an investigation of optical properties of HfTa$_4$C$_5$ and its high-entropy carbide derivatives. Results are promising: for all systems, we identify the existence of a crossover energy in the infrared and visible range, which corresponds to a metal-to-dielectric transition, exploitable for optical and telecommunication applications. The optical response of plasmonic high-entropy carbides (PHECs) can be largely tuned from the near-IR to visible by changing the composition. For a few systems, this corresponds to the excitation of a low-energy screened plasmon. HfTa$_4$C$_5$, HfNbTaTiZrC$_5$ (3), and HfNbTaTiWC$_5$ (14) exhibit the best plasmonic properties (i.e., low-energy loss and high lifetime) among the

investigated compounds. Other systems are also useful: their higher dissipative character can be used for exploiting de-excitation of plasmons and the localized release of energy, such as in photo-thermal applications. The analysis of the electronic structure allows us to identify the chemical elements (e.g., Ta, Hf, Nb, Zr) capable of optimizing the optical properties and to design high-entropy ceramics, whose optical properties are tailored to the specific application needs. The combination of plasmonic activity, high-hardness and extraordinary thermal stability, makes PHECs an example of multifunctional plasmonic-mechanical materials that can be used as optical systems in harsh environments.

## Methods

### Theory: structure generation

The macroscopic disorder is factorized as the sum of many microscopic ordered *j*-configurations which are determined by the POCC method[41], as implemented within the AFLOW computational materials design framework[57,62]. The method generates a factor-group weighted Boltzmann ensemble of Hermite normal form superlattices having the minimum size compatible with the required stoichiometry and accuracy. Non-unique supercells are removed through symmetry[58] and identification[59] considerations. HfTa$_4$C$_5$ is simulated with two sets of structures, 5 unique 10-atom cells and 77 unique 20-atoms cells, used to check the effect of the structure size on the optical properties of the systems. For each investigated PHEC system, POCC generates 49 unique 10-atom structures. The unique structures are relaxed and the total are energies calculated within the AFLOW framework with standard parameters[63], using the VASP density-functional theory package[64] with the Perdew-Burke-Ernzerhof (PBE) exchange correlation functional[65], and a grid of at least 8000 k-points per reciprocal atom.

### Theory: optical properties and ensemble integration

For each POCC configuration and for each HEC system, optical properties are evaluated in linear response by using the turboEELS code[66] included in the Quantum ESPRESSO distribution[67,68]. turboEELS implements a Liouville-Lanczos approach to linearized time-dependent density-functional perturbation theory (TD-DFPT)[69] for the evaluation of the complex susceptibility $\chi(\mathbf{q}, E)$, where $\mathbf{q}$ is transferred momentum and $E$ is the energy of the incoming radiation. The complex dielectric function $\hat{\epsilon}(E)$ and the electron-energy loss function are calculated through the relations $\epsilon(\mathbf{q}, E) = 1 + \chi(\mathbf{q}, E)$ and $EELS(\mathbf{q}, E) = -\mathrm{Im}[\epsilon^{-1}(\mathbf{q}, E)]$. The optical response function $\hat{\epsilon}(E) = \epsilon(\mathbf{0}, E)$ is obtained in the limit $\mathbf{q} \rightarrow \mathbf{0}$. The capability of the present approach in simulating the optical properties of plasmonic materials has been previously established in refs. 32, 36. As a further accuracy test, in the Supplementary Information we include a comparison (Supplementary Fig. 7) between the dielectric function of the reference TaC rocksalt crystal and of 3-HfNbTaTiZrC$_5$ high-entropy carbide, calculated with TD-DFPT and with a single-particle Drude-Lorentz approach[70] often adopted for plasmonic studies[29,34,38]. PBE is also used here. Atomic potentials are described by ultrasoft pseudo-potentials of Vanderbilt type[71]. 3s3p, 4s4p, 5s5p semicore electrons are explicitly included in the valence shell of period 4 (Ti, V, Cr), period 5 (Zr, Nb, Mo), and period 6 (Hf, Ta, W) atomic elements, respectively. Single-particle wavefunctions (charge) are expanded in plane waves up to a kinetic energy cutoff of 28 Ry (280 Ry). A uniform mesh of (12 × 12 × 12) k-points was used to sample the 3D Brillouin zone of each POCC configuration. The optical $(L, \epsilon)$ and electronic (eDOS) spectra for the PHECs are obtained along the lines of ref. 41. The ensemble average of the generic physical property $\mathcal{O}$ is given by the formula $\mathcal{O} = \sum_j P_j \mathcal{O}_j$, where $\mathcal{O}_j$ and $P_j = g_j e^{-\Delta H_j / k_B T} / \sum_j g_j e^{-\Delta H_j / k_B T}$ are the spectrum and the probability of the *j*-configuration, respectively. $\Delta H_j$ and $g_j$ are the relative formation enthalpy with respect to the multi-dimensional convex hull[72] and the symmetry degeneracy (factor-group

cardinality)[58] of the $j$-configuration, where $k_B$ is the Boltzmann constant, and $T$ is the temperature.

## Experiments: sample preparation

Tantalum carbide (99.5% purity, Stanford Advanced Materials) and hafnium carbide (99.0% purity, H.C. Starck) were blended at a 4:1 ratio, respectively, in a Nalgene plastic jar with 3/16″ WC-Co satellites and ball-milled at a 1:1 ball-to-powder ratio for 24 h. Then, bulk $HfTa_4C_5$ was sintered using field assisted sintering technology (FAST), which allows for sintering at high heating rates and short processing times without sintering aids[73]. A 25 Ton FAST system (FCT Systeme GmbH) at the Penn State Applied Research Laboratory was used for sintering the powders. The powders were sintered in a 40 mm OD graphite die to a final pellet thickness of ~4 mm. The sintering was carried out in two concurrent steps at temperatures (2100 °C/2400 °C), pressures (55 MPa/40 MPa), and hold times (40 min/30 min) and was completed at a uniform heating rate of 100 °C/min and under vacuum at ~3 mTorr. Density was measured to be 94.4% of the theoretical value using the Archimedes principle on a precision digital analytical balance (AND HM-202, ±0.1 mg).

## Experiments: EELS spectra

Cross-sectional scanning transmission electron microscopy (STEM) and EELS were performed using an aberration corrected ThermoFisher Titan[3] G2 60–300 with a monochromator and an X-field emission gun source at a beam energy of 300 keV. The $HfTa_4C_5$ specimen used for experimental EELS data acquisition was prepared according to Supplementary Fig. 9. Spectral resolutions of ≤0.2 eV were achieved for all EELS measurements as calculated by the full width at half maximum (FWHM) of the zero loss peak (ZLP). Low-loss EELS spectra were collected from $HfTa_4C_5$ as a function of temperature, ranging from room temperature (~25 °C) to 1200 °C, where the specimen was heated at a uniform rate of 10 °C/s. EELS spectra were collected after stabilizing at each target temperature. A power-law decay function was fitted to the tail of the ZLP in front of the first absorption feature in order to filter out ZLP background signal and resolve the features of interest in the spectra. This was carried out using the Gatan DigitalMicrograph software suite.

## Data availability

All the ab initio data are freely available to the public as part of the AFLOW online repository and can be accessed through AFLOW.org following the REST-API interface[74] and AFLUX search language[75]. Source data are provided with this paper.

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

## Acknowledgements

We thank Hagen Eckert, Simon Divilov, Don Brenner, Jon-Paul Maria, Bill Fahrenholtz, Eva Zurek, Adam Zettel, and Rico Friedrich for fruitful discussions. The authors acknowledge support by DOD-ONR N00014-21-1-2132, N00014-20-1-2525, N00014-20-1-2299). This work was supported in part by high-performance computer time and resources from the DoD High Performance Computing Modernization Program.

## Author contributions

A.C. and S.C. envisioned, designed and planned the project. A.C. performed the optical properties calculations; C.O., C.T., and M.E. performed the single-phase POCC simulations and obtained the disorder description. S.S. and D.W. prepared and measured the samples. All authors, A.C., C.O., C.T., M.E., X.C., S.S., D.W., and S.C., discussed the results and contributed to the writing of the article.

## Competing interests

The authors declare no competing interests.
