## [Peer Review File · Nature Communications]

Plasmonic high-entropy carbidesREVIEWER COMMENTS

Reviewer #1 (Remarks to the Author):

In this work, Calzolari et al. investigate the potential of high-entropy transition-metal carbides for applications in high-temperature plasmonic devices with computational first-principles approaches. Using thermodynamic disorder modeling in combination with time-dependent electronic structure theory, the authors demonstrate that high-entropy transition-metal carbides can be useful in such applications, especially since their optical response can be largely tuned via their composition. In particular, the authors suggest routes to design and tailor these high-entropy ceramics.

The topic of this manuscript is certainly important and timely; the ability to design high-temperature plasmonic devices certainly important for technological applications, as nicely summarized in the introduction of this work. And indeed, these materials provide an appealing “simultaneous combination of plasmonic activity, high-hardness, and extraordinary thermal stability”, as the authors highlight in the abstract. Nonetheless, I have some reservations regarding the publication of this work in Nature Communications, as summarized in the following.

High-entropy carbides are not completely novel compounds, but have been proposed by some of the authors in 2018, see Ref. [40]. Since then, their properties have been studied extensively, especially for high-temperature applications, see Refs. [42-54]. Accordingly, the fact that these compounds feature “high-hardness and extraordinary thermal stability” is no novel result, but has been substantiated in literature several times, also by some of the authors. The novelty in this manuscript is to utilize and tailor these compounds specifically for plasmonic applications by tuning their optical response. The evaluation of these optical properties appears to be, however, not particularly rigorous – at least from what I can infer from the given description.

(a) The employed Partial Occupation (POCC) algorithm used to model disorder relies on the idea that the effect of disorder can be factorized as the sum over Boltzmann-weighted microscopic configurations, in the spirit of an ensemble average. This approach has been used successfully in previous studies to model thermodynamic expectation values, for which such an ensemble average approach is obviously justified. To my knowledge, the POCC approach has however so far not been used to model electronic response properties as it is the case here. Since such properties explicitly rely on the time-dependent response of the system, it is yet unclear to which extent an “ensemble average” approach is fully justified, a reasonable approximation, or maybe questionable. Here, I feel that an explicit discussion and validation of the chosen methodology for modeling the combination of disorder and plasmonic response is necessary to substantiate the predictions.

(b) Furthermore, the main novelty in this manuscript revolves around using these high-entropy transition metal carbides in high-temperature applications. Here, the authors claim that the “plasmon resonance remains incredibly stable even at ultra-high temperature”. This claim is however only based on “adjusting the temperature used to average the POCC ensemble”, a very crude model. All other thermodynamic effects like electronic temperature, nuclear motion, thermodynamic reorganization of the electronic structure, and electron-phonon coupling appear to be neglected without further justification. Here, a more rigorous investigation of thermodynamic effects and their potential influence on the predictions seems necessary.

Reviewer #2 (Remarks to the Author):

The authors present a computational study showing how high corrosion and damage resistance can be combined with attractive plasmonics properties in multi-component refractory metal carbides. The study is of potential interest for Nature Communications granted that the proposed material truly outperforms other options. The manuscript will require some changes to clarify a few scientific and technical issues but also to make sure the presentation appeals to a broader audience.

In terms of presentation, I would recommend for the authors to revise their manuscript to appeal to a broader audience. While the introduction is good, the main text/results should remind the reader what are the real optical properties that are essential for a high performance plasmonic material. They should also mention properties the authors might not have been able to compute. The discussion on the chemical and composition effects is hard to follow and the presentation would benefit from a figure summarizing the results and showing trends let say across the periodic table. Finally, the author should clarify for a broad audience why EELS spectra are important in this field. The transition from dielectric to EELS is not clearly explained.

For the scientific part, there are some statements that should be backed-up and clarified. The authors argue that their proposed material have exceptional mechanical properties. Do they have anything to back this up from theory or experiment? See p. 5 "HECs have exceptional mechanical properties". The authors also show that the plasmon resonance remains stable at high temperature but they do not model effects such as vibrations that could also play a role at higher temperature. If they were running the same simulation at high temperature on let say Ag, would they see a stable spectra or not?

Losses seem an important problem in plasmonic materials. These losses can be due to scattering from phonons, impurity etc... Do the authors model this? How important are those factors in predicting plasmonic properties? This needs to be clarified to judge how predictive the approach is.

A few more technical points need to be addressed as well:

- The authors rely on TD-DFT to compute EELS and dielectric constant. Previous work from the authors in refs 38 used an independent particle approach. The author should justify why using one or the other.

- The authors assume that their EELS/dielectric constant spectra are well converged by their POOC algorithm. As far as I know, these types of methods are mainly used for quantities such as energies. The authors should show evidences that the POOC approach indeed recovers the optical spectra of a fully disordered system.

Finally, but this is a matter of editorial decision. The materials described have been made and reported in Refs 40, 44. Samples of HfNbTaTiZrC₅ and HfNbTaTiWC₅ are available to the authors. It is surprising to not have any experimental optical data to at least show the predicted plasmon. This seems to me not a very difficult experiment to make.

Reviewer #3 (Remarks to the Author):

Calzolari and co-authors investigate the optical properties of HfTa₄C₅ and its high-entropy carbide derivatives, namely, MTa₄C₅, where M = Ti, Zr, Hf, V, Nb, Cr, Mo, W, using time-dependent density functional theory. They use crossover energy in the infrared and visible range and EELS to identify optimal carbide composition. They found that HfTa₄C₅, HfNbTaTiZrC₅, and HfNbTaTiWC₅ exhibit the best plasmonic properties (i.e., low energy loss and high lifetime). This study is of importance for applications because these high entropy carbides combine the properties of plasmonic activity, high-hardness and extraordinary thermal stability. The paper is written, and approach is novel, and findings are new and important. Therefore, the reviewer recommends accept as it is.

Minor: Page 1, define CMOS please.

Reviewer #1 (Comments to the Authors)

Q1.1. In this work, Calzolari et al. investigate the potential of high-entropy transition-metal carbides for applications in high-temperature plasmonic devices with computational first-principles approaches. Using thermodynamic disorder modeling in combination with time-dependent electronic structure theory, the authors demonstrate that high-entropy transition-metal carbides can be useful in such applications, especially since their optical response can be largely tuned via their composition. In particular, the authors suggest routes to design and tailor these high-entropy ceramics.

*The topic of this manuscript is certainly important and timely; the ability to design high-temperature plasmonic devices certainly important for technological applications, as nicely summarized in the introduction of this work. And indeed, these materials provide an appealing “simultaneous combination of plasmonic activity, high-hardness, and extraordinary thermal stability”, as the authors highlight in the abstract. Nonetheless, I have some reservations regarding the publication of this work in *Nature Communications*, as summarized in the following.*

Authors:

A1.1. We thank the reviewer for their assessment of our work.

Reviewer #1 (Comments to the Authors)

Q1.2. High-entropy carbides are not completely novel compounds, but have been proposed by some of the authors in 2018, see Ref. [40]. Since then, their properties have been studied extensively, especially for high-temperature applications, see Refs. [42-54]. Accordingly, the fact that these compounds feature “high-hardness and extraordinary thermal stability” is no novel result, but has been substantiated in literature several times, also by some of the authors. The novelty in this manuscript is to utilize and tailor these compounds specifically for plasmonic applications by tuning their optical response. The evaluation of these optical properties appears to be, however, not particularly rigorous – at least from what I can infer from the given description.

Authors:

A1.2. While the extreme thermo-mechanical properties of high entropy carbides are, by themselves, not a novelty, the discovery of their plasmonic properties, is nonetheless novel. Standard plasmonic systems, namely Au, Ag, Cu, are ductile metals with low melting temperature. Even in alloyed form, they hardly allow for tailoring of optical and plasmonic properties which hinders applications. The identification of a class of multifunctional materials combining mechanical, thermal and plasmonic properties is as relevant as the whole is greater than the sum of its parts: the connection of two separated research realms (i.e. structure versus and optical) opens to novel potential optical-mechanical applications.

The optical properties of the systems have been evaluated by using a state-of-the-art technique, based on a Liouville-Lanczos approach to linearized time-dependent density-functional theory [Phys. Rev. B 88, 064301 (2013)]. The method allows for a fully ab initio description of the non-local, frequency-dependent microscopic susceptibility $\chi(\mathbf{q}, \omega)$, the complex dielectric function $\epsilon(\mathbf{q}, \omega) = 1 + \chi(\mathbf{q}, \omega)$, and the whole optical properties (including the electron energy loss function $EELS(\mathbf{q}, \omega) = -\text{Im}[\epsilon^{-1}(\mathbf{q}, \omega)]$). The adopted approach includes crystal local field effects and exchange-correlation local field effects [Phys. Rev. B 81, 085104 (2010)] that account for non-homogeneity of the system on the microscopic scale. This method treats with the same accuracy both interband and intraband contributions to electronic excitations; it does not include electron-phonon scattering terms, which could be relevant in the low-energy limit ($\omega \rightarrow 0$), an energy range not considered here. Excitonic effects, which are best described by many-body approaches (e.g. GW-Bethe Salpeter), are not relevant for the investigated metallic systems.

The identification of the crossover energy and of the plasmon excitations has been done on the basis of the poles of the complex dielectric function. The poles of the dielectric functions correspond to peaks in the loss function, which can be directly compared with EELS experimental data, as previously done by some of the authors [Phys. Rev. B 95, 115145 (2017)] by comparing simulated and the experimental EELS spectra of TiN (twin plasmonic material). The published comparison is presented below.

FIG. 2. Simulated loss function for TiN bulk at increased transferred momentum $|\mathbf{q}|$ in units of $2\pi/a_0$. Dot-dashed line follows the low-energy single-particle ($el-h$) contribution at increasing $|\mathbf{q}|$. Two sets of experimental EELS spectra (gray curves) are superimposed for comparison. Experimental set 1 is adapted from Ref. [59], and set 2 is from Ref. [58].

Image adapted from Phys. Rev. B 95, 115145 (2017)

For extended bulk systems, such as the ones addressed in the submitted article, no further morphological or size parameters are necessary to identify the plasmonic resonance.

To better explain how plasmonic features are evaluated, we have included a sentence in the main text (page2): *“The response of materials to incoming electromagnetic waves is described by the complex dielectric function $\epsilon(q, E) \equiv (\epsilon_r + i\epsilon_i)$, where q is the transferred momentum and E is the energy of the field [55]. The real part, ϵ_r , relates to polarization effects and distinguishes between dielectrics ($\epsilon_r > 0$) and metals ($\epsilon_r < 0$). The imaginary part, ϵ_i , is associated with dissipation of energy into the medium. For bulk systems, the electron energy loss spectra (EELS)*

is defined as $-\text{Im}[\epsilon^{-1}(q,E)]$ and it describes the energy loss due to inelastic electron scattering upon electromagnetic irradiation. In the low-loss energy region ($E < 50$ eV), the spectrum of EELS provides information about electronic structure, optical, and plasmonic properties. The plasmon excitation energies are the poles of $\epsilon(q,E)$ and correspond to EELS's peaks. Experimentally, plasmons are determined by measuring the characteristic electron-energy loss. The evaluation of $\epsilon(q, E)$ and the corresponding EELS completely characterizes the plasmonic features, and provides a direct pathway to comparison with experimental results [32]. ”.

Reviewer #1 (Comments to the Authors)

Q1.3. (a) The employed Partial Occupation (POCC) algorithm used to model disorder relies on the idea that the effect of disorder can be factorized as the sum over Boltzmann-weighted microscopic configurations, in the spirit of an ensemble average. This approach has been used successfully in previous studies to model thermodynamic expectation values, for which such an ensemble average approach is obviously justified. To my knowledge, the POCC approach has however so far not been used to model electronic response properties as it is the case here. Since such properties explicitly rely on the time-dependent response of the system, it is yet unclear to which extent an “ensemble average” approach is fully justified, a reasonable approximation, or maybe questionable. Here, I feel that an explicit discussion and validation of the chosen methodology for modeling the combination of disorder and plasmonic response is necessary to substantiate the predictions.

Authors:

A1.3. The partial occupation approach (POCC) has previously used with success to characterize ensemble averages of atomic configurations, electron and phonon density of states, both ground-state properties or thermodynamic excited states for the system. Here we extend POCC to electronic excited states (i.e. optical properties). POCC is expected to well reproduce all disorder properties which have characteristic lengths larger than the one of the radial distribution function, which is captured by the size cut-off of the POCC cells. As long as the plasmonic excitation has a size which contains enough statistical representatives of the solid-solution chemical disorder, the ensemble will provide a good description of the excitation.

However, only a test can settle this expectation. As accuracy testbed, we compare our results to the archetype of plasmonic materials, the disordered solid-solution AuAg and the fcc constituents Au and Ag. As shown in Figure S3 of the Supplementary Information, our approach accurately reproduces the dielectric function of the disordered material that has specific optical features, different from the corresponding crystalline version, as proved by the comparison with experiments. We also investigate the effect of the configurational temperature on the optical properties of disordered systems, which is known to be related to structural thermal contraction/expansion of the material (see below for a further analysis of effect of electron-phonon interaction of plasmon properties).

This confirms that, at least for this class of metallic systems, the proposed approach accurately characterizes the effects of disorder on the optical and plasmonic properties of materials.

The new set of results on AuAg and the corresponding discussion has been explicitly included in the main text which now reads (page2) “*The capability of this approach in reproducing the electron and phonon density of states has already been proved [60]. Here we extend it to optical spectra of disordered systems. For each optimized POCC j-structure, we calculate $\hat{\epsilon}(q, E) \equiv (\epsilon + i\epsilon)$, and the corresponding EELS, $-Im[\epsilon^{-1}(q, E)]$. Unless otherwise specified, we consider j only the optical limit at $q=0$. Then, the spectra of the disordered material are obtained as an ensemble average on the POCC j-structures (see Methods Section). The poles (peaks) of the averaged dielectric function (EELS) defines the spectral position of the plasmon resonance. As accuracy testbed, we compare our results to the archetype of plasmonic materials, the disordered solid-solution AuAg and the fcc constituents Au and Ag. The results are summarized in Figure S3 of the Supplementary Information. The excellent agreement between simulated and experimental findings confirms the accuracy of the POCC method in calculating optical properties of disordered systems. This is not surprising: POCC is expected to well reproduce all disorder properties which have characteristic lengths larger than the one of the radial distribution function, which is captured by the size cut-off of the POCC cells. On the basis of these results on well-known plasmonic materials, we use this method to predict the plasmonic properties of high entropy carbides.*”.

The Supporting Information now include (page S3): “*Here we test the capability of the POCC method in simulating the optical spectra of disordered systems. We consider the case of the disordered AuAg solid-solution, compared to the Au and Ag fcc constituents, all well-known plasmonic systems. AuAg is simulated by using 44 POCC structures with 8 atoms per cell. The imaginary part ϵ_i of the dielectric function of the three systems is shown in Figure S3(a), along with the corresponding experimental counterpart (inset), extracted from Ref. [1]. The disordered alloy has a distinct character, with spectral features not directly ascribable to the ordered crystalline phases of the single elements. The excellent agreement with the experimental findings confirms the accuracy of the present approach in including the effect of disorder into the description of the optical properties.*”.

Figure S3. a) Imaginary part (ϵ_i) of the complex dielectric function of Au (blue), Ag (red) and AuAg (yellow) systems. Inset reports the corresponding experimental spectra as extracted from Figure 3(b) of Ref. [1]. b) Simulated Electron Energy Loss spectrum (EELS) of disordered AuAg solid-solution at different configurational temperature.”

Reviewer #1 (Comments to the Authors)

Q1.4. (b) Furthermore, the main novelty in this manuscript revolves around using these high-entropy transition metal carbides in high-temperature applications. Here, the authors claim that the “plasmon resonance remains incredibly stable even at ultra-high temperature”. This claim is however only based on “adjusting the temperature used to average the POCC ensemble”, a very crude model. All other thermodynamic effects like electronic temperature, nuclear motion, thermodynamic reorganization of the electronic structure, and electron-phonon coupling appear to be neglected without further justification. Here, a more rigorous investigation of thermodynamic effects and their potential influence on the predictions seems necessary.

Authors:

A1.4. We thank the referee for pointing out these issues.

Thermal effects. The structural and vibrational entropy associated to the temperature within the POCC algorithm have been explored in previous articles cited here. While lattice vibrations contribute to vibrational entropy, the latter is expected to be smaller than the configurational part for the rocksalt class of systems we investigate, where most of the constituents and sub-components are also rocksalt [Nature Communications 12, 5747 (2021)]. Therefore, the temperature used inside the POCC ensemble is a good representation of the real temperature of the material during synthesis.

Plasmonic effects. The effect of temperature on plasmon excitation is more complicated. First, it is necessary to distinguish between the effects on the **1)** excitation and on the **2)** de-excitation of the plasmon resonance.

1) Excitation. The excitation of plasmons is a purely electronic effect (intraband electron-electron excitation) and it is related to the electron temperature of the electron gas (several tens of thousands of K). The main loss effects are due to intraband transitions that are responsible for the contributions to the imaginary part of the dielectric function (optical absorption). The excitation of the electron gas is an ultrafast process (on the fs timescale) that does not involve electron-phonon interaction. While hot-electron excitations have been demonstrated to shift the energy position of the plasmonic band, they correspond to ultrafast changes of the electron temperature, e.g., to pump flux in pump-probe experiments. The “ultra-high-temperature” range considered in this work ($T < 3000\text{K}$) is still much lower than the corresponding electron temperature and therefore we can consider the electrons responding instantaneously to slowly moving ions (Born–Oppenheimer approximation). Conversely, the ionic temperature has an indirect and minor effect of the plasmon excitation energies, mostly due to the thermal contraction/expansion of the lattice. As shown in Figure S3b of SI, this effect is correctly described by the POCC approach, which controls the spatial distribution of the atoms in the disordered system. Beyond a certain temperature (system dependent), structural instabilities and/or phase transitions (e.g. melting) will change the morphology of the system, yet affecting optical properties. This is not considered in the article which deals with thermodynamically homogeneous systems, and therefore, we can conclude that away from critical temperatures, temperature has a secondary effect on the plasmonic excitation energy and that our approach well describes this process.

2) *De-excitation*. The stability (i.e. lifetime) and of the de-excitation of the plasmonic resonance follows a different scenario. Here, the interaction with the lattice, and thus with the ionic temperature, is critical. After the initial ultrafast thermalization of the hot-electron gas through electron-electron scattering, the energy is transferred to the lattice via phonon emission (electron-phonon coupling) on the nanosecond scale. Finally, thermal energy is radiated (heat-diffusion) towards the external environment through phonon-phonon scattering on the micro-to-millisecond scale. While increasing temperature may favor the plasmon extinction through destructive electron-phonon interaction, it does not directly affect the energy position of the plasmon resonance and/or of the crossover energy (beyond considerations about thermal expansion).

To clarify this critical fact, we have changed the main text by including a new paragraph on the effects of temperature and the interpretation of the presented results. The text reads (page 6): “*Since plasmons are purely electronic excitations, ionic temperature does not affect the energy position of the plasmonic band (i.e., the crossover energy), except for effects related to thermal expansion of the atomic structure [55] (see the Supplementary Information for further details). The results of Figure 4(d) indicate the possibility to excite a plasmon resonance even at ultrahigh-temperature, where structural resistant of HECs can be advantageous for applications. Temperature could still affect lifetime and de-excitation of the plasmons through dissipative electron-phonon scattering. Our approach does not include time evolution — de-excitation and related energy release — which is beyond the purpose of this work.*”

Further discussion and complementary data are included in Supplementary Information. The text reads (page S3). “*The excitation of plasmons is a purely electronic effect, i.e., intraband electron-electron excitations. The excitation of the electron gas is an ultrafast process (on the femtosecond scale) that does not involve the interaction with phonons. The ionic temperature has an indirect and minor effect of the plasmon excitation energies, mostly ascribable to the thermal expansion of the lattice and thus to the average modifications of the bond lengths. Figure S3(b) shows the simulated EELS spectra of disordered AuAg solid-solution at different configurational temperatures. The results match well with the measured redshift of the plasmonic band (see gray dashed line) of AuAg as the temperature is increased [2]. This effect, usually of the order of 0.1 eV, has been observed also for Ag bulk systems and it is associated to thermal structural expansion [3]. As shown in Figure 1(b), this effect is reproduced by POCC, which describes the spatial distribution of the atoms in the disordered system. Beyond a certain temperature (system dependent), structural instabilities and/or phase transitions (e.g., melting), will change the morphology of the system, yet affecting optical properties. This is not considered in the article because it focuses on thermodynamically homogeneous systems. Therefore, we can conclude that away from critical temperatures, temperature has a secondary effect on the plasmonic excitation energy and that our approach well describes this process.*

The stability (i.e., lifetime) and of the de-excitation of the plasmonic resonance follows a different scenario [4, 5]. Here, the interaction with the lattice, and thus with the ionic temperature, is critical. After the initial ultrafast thermalization of the hot-electron gas through electron-electron scattering, the energy is transferred to the lattice via phonon emission (electron-phonon coupling) on the nanosecond scale. Finally, thermal energy is radiated (heat-diffusion) towards the external environment through phonon-phonon scattering on the micro-to-millisecond scale. While increasing temperature may favor the plasmon extinction through destructive electron-phonon

interaction, it does not directly affect the energy position of the plasmon resonance and/or of the crossover energy (beyond considerations about thermal expansion). Therefore, we can conclude that below the critical melting point, temperature has a minor effect on the plasmonic excitation energy and that the POCC method well characterizes the phenomenon.”

Reviewer #2 (Comments to the Authors)

Q2.1. The authors present a computational study showing how high corrosion and damage resistance can be combined with attractive plasmonics properties in multi-component refractory metal carbides.

The study is of potential interest for Nature Communications granted that the proposed material truly outperforms other options. The manuscript will require some changes to clarify a few scientific and technical issues but also to make sure the presentation appeals to a broader audience.

Authors:

A2.1. We thank the reviewer for their constructive criticism and their favorable remarks at the end.

Reviewer #2 (Comments to the Authors)

Q2.2. In terms of presentation, I would recommend for the authors to revise their manuscript to appeal to a broader audience. While the introduction is good, the main text/results should remind the reader what are the real optical properties that are essential for a high performance plasmonic material. They should also mention properties the authors might not have been able to compute. The discussion on the chemical and composition effects is hard to follow and the presentation would benefit from a figure summarizing the results and showing trends let say across the periodic table. Finally, the author should clarify for a broad audience why EELS spectra are important in this field. The transition from dielectric to EELS is not clearly explained.

Authors:

A2.2. We agree with the reviewer: the broad audience deserves a more accessible presentation. We have performed an overall revision of the text. In particular, we have included a new paragraph at the beginning of the “Results” Section. There, we introduce and describe the main optical descriptors, the approaches used to evaluate them, and comparison with the experimental counterpart.

While we agree with the suggestion of including a new Figure summarizing the trends across the periodic Table, we recognize the impracticality of representing physical properties (e.g. E_0 , h) as scalar functions of 5 independent variables (i.e. the transition-metal contents of the compounds), which could be plot in a 3D diagram, then printed in a page. In addition, given the limited amount of compositions that we have investigated, points would be completely scattered. Thus, we feel

that the task is currently unsolvable, and we would prefer not to change the graphical render of the results.

To improve the readability of the manuscript we rephrased the section of chemical comparison which now reads (pages 3-4): “*Figure 3(a) shows the energy variation of E_0 as a function of composition. The systems are grouped into four classes with respect to the TM groups (Table 1). The first class, 3g4 in Figure 3, maximizes the number of group 4 elements (Ti, Zr, Hf); the second, 3g5, maximizes the group 5 metals (V, Nb, Ta); the third, 3g6, those of group 6 (Cr, Mo, W), with the remaining two elements per systems are from the other groups; the last (2g42g5) has a mixed composition, with two elements from group 4 and 2 from group 5. Figure 3(a) shows some evident trends. Wide ranges. E_0 spans a remarkable range of energies from near-IR to visible. This is quite interesting, as the possibility to control and tune the optical properties of materials by varying the composition is of critical relevance in the development of plasmonics and nanophotonics applications. Intragroup increase. Within each group, E_0 increases monotonically with the increase of the atomic number Z of the constituents. For example, systems 1-3 have in common three elements of group 4 (Ti, Zr, Hf) and differ for the remaining two elements from the group 5, V-Nb (1), V-Ta (2), and Nb-Ta (3). Indeed, E_0 increases from system-1 to 3 along with the atomic number of the constituents. Intergroup increase. Across different groups, the higher the Z of the elements, the higher becomes the crossover energy. E_0 tends to increase moving from 3g4 to 3g6, i.e., increasing number of electrons. For mixed compositions, e.g., system 14-HfNbTaTiWC5, E_0 spans the intermediate energy range.”*

Reviewer #2 (Comments to the Authors)

Q2.3. *For the scientific part, there are some statements that should be backed-up and clarified. The authors argue that their proposed material have exceptional mechanical properties. Do they have anything to back this up from theory or experiment? See p. 5 "HECs have exceptional mechanical properties". The authors also show that the plasmon resonance remains stable at high temperature but they do not model effects such as vibrations that could also play a role at higher temperature. If they were running the same simulation at high temperature on let say Ag, would they see a stable spectra or not?*

Authors:

A2.3. We thank the referee for pointing out these issues. The mechanical, vibrational and thermal properties of high entropy carbides have been extensively studied both theoretically (mostly by some of the authors of the present work) and experimentally [*Adv. Mater.* 202102904 (2021), *Acta Mater.* **166**, 271-280 (2019), *Nature Communications* **9**, 4980 (2018)]. As the thermal stability of plasmon is concerned, here we focused on the possible modifications, in terms of energy shift and intensity reduction, of the plasmonic band. As discussed above, the ionic temperature has a minor effect on the energy of the plasmon resonances, while it could have a major in the lifetime and de-excitation process of the plasmon, which however are not considered in this work. Accuracy tests on the effect of temperature on AuAg solid-solution have been included in the Supplementary Information, along with the comparison with the experimental results.

In order to clarify this important aspect, we changed the main text and included a new paragraph on the effects of temperature and the interpretation of the presented results. Further discussion and complementary data are included in Supplementary Information, for comprehensiveness. Please see answer **A1.4** to Reviewer 1 question.

Reviewer #2 (Comments to the Authors)

Q2.4. Losses seem an important problem in plasmonic materials. These losses can be due to scattering from phonons, impurity etc... Do the authors model this? How important are those factors in predicting plasmonic properties? This needs to be clarified to judge how predictive the approach is.

Authors:

A2.4. The Referee is correct: losses are important problems in plasmonics. Yet, our article focuses on the discovery of plasmonic features in a class of materials which was not known to have plasmonic properties, and we believe this novelty to be of interest for the general audience. We demonstrate the possibility to excite a plasmon resonance in the near-IR/visible range, and we do not consider the time evolution of the plasmon quasiparticle and its de-excitation process. Thus, only the electronic energy loss (i.e. interband transition) are included in this model, as the description of the coupling with phonons or impurities goes beyond the purpose of the report. In addition, the role of the loss effects crucially depends on the final applications: while low-loss and long lifetimes are crucial for applications that exploits the existence of the plasmonic excitation (e.g. waveguides), high loss and short lifetime are desired for applications that exploit the de-excitation energy release and thermal effects. The optimization of materials for specific application will be the subject of forthcoming investigations.

Reviewer #2 (Comments to the Authors)

*Q2.5. A few more technical points need to be addressed as well:
-The authors rely on TD-DFT to compute EELS and dielectric constant. Previous work from the authors in refs 38 used an independent particle approach. The author should justify why using one or the other.*

Authors:

A2.5. The reason of this choice is technical: the class of materials and the properties of interest dictate the most efficient solution. In previous works [Phys. Rev. B 95, 115145 (2017), ACS Photonics 5, 2816 (2018)], we have demonstrated for the twin system TiN, that the two approaches, TD-DFPT and single particle (SP) Drude-Lorentz, give the same optical dielectric function $\hat{\epsilon}(E) = \hat{\epsilon}(\mathbf{0}, E) = \epsilon_r + i \epsilon_i$, same real and imaginary part of the dielectric function at zero transferred momentum ($\mathbf{q}=\mathbf{0}$), and thus the same crossover energy $E_o(\mathbf{0})$ and the same loss function $EELS(\mathbf{0}, E) = -\text{Im}[\epsilon^{-1}(\mathbf{0}, E)]$. In Reference [38], we considered the combination of the optical properties of the entire set of rocksalt TMX compounds, where TM were the 30 transition metals

and $X=B, N, C$, with the aim of investigating the possible hyperbolic metamaterial behavior of rocksalt-based superlattices in the near-IR and visible range. There, a few systems (ScN, YN, and LaN) were semiconductors, with small indirect bandgaps ($E_g < 1.0$ eV). Considering the well-known bandgap underestimation of standard DFT-GGA, and to obtain the correct interband optical edge, we applied a DFT+ U approach [Phys. Rev. X **5**, 011006 (2015); ACS Photonics **1**, 703 (2014); Opt. Exp. **26**, 5342 (2018)]. Currently, the DFT+ U approach for the evaluation of the optical properties is implemented only in the SP approach within the Quantum ESPRESSO suite of codes, which we used for calculations (i.e. there is no TD-DFPT+ U as of now).

Given that our HECs are all metallic, we do not need to deal with bandgap corrections, and we are only interested in the dispersion of the crossover energy versus the transferred momentum, $E_c(\mathbf{q})$ - the fingerprint of the plasmonic character of the resonance (Figure 4 main text). The non-local complex dielectric function $\hat{\epsilon}(\mathbf{q}, E)$ and the corresponding loss function $EELS(\mathbf{q}, E)$ are available only through the TD-DFPT implementation of the Quantum ESPRESSO code. This explains the choices of the two different methods. While equivalent in the case of the optical ($\mathbf{q}=0$) properties of metals, the actual code implementation dictates the approach to follow in the different cases.

To avoid confusion, we modified the submission in two places.

In the **main text**, we have added a sentence in the methodological section which summarizes the similarities and the differences of the two approaches and the reason on the actual TD-DFPT choice which reads as (page 7): “*The capability of the present approach in simulating the optical properties of plasmonic materials has been previously established in Refs. [32, 36]. As a further accuracy test, in the Supplementary Information we include a comparison (Figure S7) between the dielectric function of the reference TaC rocksalt crystal and of 3-HfNbTaTiZrC₅ high-entropy carbide, calculated with TD-DFPT and with a single particle Drude-Lorentz approach [69] often adopted for plasmonic studies [29, 34, 38].*”

In the **Supplementary Information**, we have included a new Figure S7 illustrating the comparison of the complex dielectric functions obtained with TD-DFPT and SP approaches in the specific cases of the reference TaC rocksalt (panel a), and of 3-HfNbTaTiZrC₅ PHEC (panel b, single POCC) (now Figure S7). The figure is reported also below. In both cases the agreement is excellent, confirming the equivalence of the two approaches in the simulation the optical response of these metallic systems.

Figure. Comparison between the optical complex dielectric function $\hat{\epsilon}(E) \equiv (\epsilon_r + i\epsilon_i)$ calculated with the Liouville-Lanczos TD-DFPT approach and the Drude-Lorentz single particle (SP) approach, as implemented in the Quantum ESPRESSO suite of codes for a) fcc TaC rocksalt crystal, and b) 3-HfNbTaTiZrC₅ high entropy carbide (single pocc). ϵ_r , ϵ_i , and E_0 identify the real, (imaginary) part of the dielectric function and the crossover energy, respectively.

Reviewer #2 (Comments to the Authors)

Q2.6. -The authors assume that their EELS/dielectric constant spectra are well converged by their POOC algorithm. As far as I know, these types of methods are mainly used for quantities such as energies. The authors should show evidences that the POOC approach indeed recovers the optical spectra of a fully disordered system.

Authors:

A2.6. As discussed above (Referee 1, **A1.3**), we demonstrated the possibility of the present approach in evaluating the optical properties of materials, by simulating the dielectric function and the EELS spectra for disordered AuAg solid solution, which has been largely investigated as plasmonic materials. We compared the results with the case of crystalline Au and Ag systems. The excellent agreement with the experimental counterpart confirms the accuracy of the present approach for the simulation of the optical properties of disordered systems.

The new set of results on AuAg and the corresponding discussion has been explicitly included in the main text and in the Supporting Information. Please see answer “**A1.3**” above.

In addition, to demonstrate the possible effect of cell size effect due to POCC we calculated the optical properties of HfTa₄C₅ by doubling the number of atoms per cell and increasing the number of POCC structures. The results, now reported in Figure 1b of main text (see below), confirms the accuracy of the presented results.

Figure 1. Optical properties of plasmonic HfTa_4C_5 : **a)** real (ϵ_r , dark gray) and imaginary (ϵ_i , light purple) part of the complex dielectric function. E_0 and E_p indicate the crossover energy and the plasmon energy, respectively. Inset zooms on low energy range of the spectrum. **b)** Simulated Electron Energy Loss spectrum (EELS) calculated by assuming two sets of structures with 10 (black) or 20 (cyan) atoms per POCC cell. Simulated EELS of TaC crystal (light gray) is included in Inset for comparison.

A sentence has been added in the main to explain the result. It reads (page 3): “*The size and the number of the adopted POCC structures do not affect this result. The simulation of disordered HfTa_4C_5 with a double number of atoms per cell (cyan line in Figure 1(b)) reproduces all the spectral features of the original system (black line) with, e.g., a difference in the crossover energy E_0 smaller than 0.15 eV.*”

Reviewer #2 (Comments to the Authors)

Q2.7. Finally, but this is a matter of editorial decision. The materials described have been made and reported in Refs 40, 44. Samples of HfNbTaTiZrC_5 and HfNbTaTiWC_5 are available to the authors. It is surprising to not have any experimental optical data to at least show the predicted plasmon. This seems to me not a very difficult experiment to make.

Authors:

A2.7. We are aware of the importance of the experiments, and it is our firm intention to extend this study and to have an experimental validation of the results. However, besides the experimental characterization of the single systems that is indeed mandatory for any actual application, here we pointed the attention on the possibility to extend the class of plasmonic materials to systems with remarkable mechanical and thermal stability. Thus, the urgency of the present work relies on the proposition of a new class of multifunctional compounds which connects the realms of plasmonics and of mechanics. For this reason, we decided to present, as a first paper on this topic, a purely theoretical investigation, postponing the experimental validation to further works.

Reviewer #3 (Comments to the Authors)

Q3.1. Calzolari and co-authors investigate the optical properties of HfTa_4C_5 and its high-entropy carbide derivatives, namely, MTa_4C_5 , where $M = \text{Ti, Zr, Hf, V, Nb, Cr, Mo, W}$, using time-dependent density functional theory. They use crossover energy in the infrared and visible range and EELS to identify optimal carbide composition. They found that HfTa_4C_5 , HfNbTaTiZrC_5 , and HfNbTaTiWC_5 exhibit the best plasmonic properties (i.e., low energy loss and high lifetime). This

study is of importance for applications because these high entropy carbides combine the properties of plasmonic activity, high-hardness and extraordinary thermal stability. The paper is written, and approach is novel, and findings are new and important. Therefore, the reviewer recommends accept as it is.

Authors:

A3.1. We thank the Referee for her/his very positive comments.

Reviewer #3 (Comments to the Authors)

Q3.2. *Minor: Page 1, define CMOS please.*

Authors:

A3.2. Thank you for pointing the issue. We have defined the acronym.

END OF REPORT

REVIEWER COMMENTS

Reviewer #1 (Remarks to the Author):

Dear Editors and Authors,

Thank you very much for addressing my comments and providing so much additional data and information. All my concerns have been addressed and resolved, so that I support publication of this manuscript in its current form.

Best wishes and apologies for the delay in reviewing.

Reviewer #2 (Remarks to the Author):

The paper has improved with the proposed revision. In view of the information provided by the authors, I believe it is not possible to know if the material would really have good plasmonic properties at high temperature (as this has not and probably cannot be modeled). Some experimental evidence on these known compounds would have helped clarifying this since modeling of this dependence with temperature seems limited at this stage. The dependence of the optical response with temperature is somehow the weak point of this overall good paper. If this dependence with temperature is not deemed critical by the editor, the paper could be published in Nature Communications. If it's critical, some experimental evidence might be needed. I note that the authors did not provide a very strong argument for not testing experimentally their prediction on a material they have published about, have access to samples and with experiments (EELS) that should not be very difficult to conduct.

Reviewer #3 (Remarks to the Author):

The authors have addressed the review comments satisfactorily, therefore, I recommend Accept.

Reviewer #1 (Comments to the Authors)

Q1. Thank you very much for addressing my comments and providing so much additional data and information. All my concerns have been addressed and resolved, so that I support publication of this manuscript in its current form.

Authors:

A1. We thank the reviewer for their assessment of our work.

Reviewer #2 (Comments to the Authors)

*Q2. The paper has improved with the proposed revision. In view of the information provided by the authors, I believe it is not possible to know if the material would really have good plasmonic properties at high temperature (as this has not and probably cannot be modeled). Some experimental evidence on these known compounds would have helped clarifying this since modeling of this dependence with temperature seems limited at this stage. The dependence of the optical response with temperature is somehow the weak point of this overall good paper. If this dependence with temperature is not deemed critical by the editor, the paper could be published in *Nature Communications*. If it's critical, some experimental evidence might be needed. I note that the authors did not provide a very strong argument for not testing experimentally their prediction on a material they have published about, have access to samples and with experiments (EELS) that should not be very difficult to conduct.*

Authors:

A2. The Curtarolo group does not have access to the samples prepared for the “2018 *Nature Communications*” article reporting the discovery of the novel carbides; the expertise of our team is theoretical/computational without experimental capabilities.

Nevertheless, to comply with the requests of the referee and editor, we managed to team up with the experimental group of Prof. Douglas Wolfe at Penn State University. The archetype carbide HfTa_4C_5 was prepared, and its EELS response was characterized from 300K to 1500K. We are happy to confirm that our plasmonic predictions are verified. Due to the extended work, the article requires the inclusion of three new authors. The appropriate paperwork for authorship-modification will be prepared once requested by the Editor.

The article has been modified as follow. Old text in black, new text in blue.

In the “Abstract”.

..... By monitoring the electronic structures, we suggest rules for optimizing optical properties and designing tailored high-entropy ceramics. Experiments performed on the archetype \$\text{HfTa}_4\text{C}_5\$ yielded plasmonic properties from room temperature to 1500K. Thus, we propose plasmonic transition-metal high-entropy carbides (PHECs) as a new class of multifunctional materials....

Inside “results”

Page3

To corroborate our claims, a set of experimental EELS measurements on the archetype high-temperature carbide HfTa_4C_5 was performed from room temperature to 1200°C (Methods and Supplementary Information). Figure 1(c) compares the simulated (dark gray) and the experimental (green) loss functions. The two curves concur (except for minor details), reproducing the main optical features of the material, in particular the screened plasmon with its maximum at $E_0 = 2.7$ eV—very close to the simulated value. In addition, the experimental spectrum has similar features to plasmonic TiN, characterized by the same EELS techniques [61]. The theoretical spectra also reproduces the low-intensity shoulder (inset, vertical arrow) close to the main E_0 peak.

Page4 – fig1c is new

Figure 1. Optical properties of plasmonic HfTa_4C_5 : **a)** real (ϵ_r , dark gray) and imaginary (ϵ_i , light purple) part of the complex dielectric function. E_0 and E_p indicate the crossover energy and the plasmon energy, respectively. Inset zooms on low energy range of the spectrum. **b)** Simulated Electron Energy Loss spectrum (EELS) calculated by assuming two sets of structures with 10 (black) or 20 (cyan) atoms per POCC cell at $T=0\text{K}$. Simulated EELS of TaC crystal (light gray) is included in inset for comparison. **c)** Comparison between simulated (dark gray) and experimental (green) EELS spectra. Inset zooms on low energy range of the spectrum.

Page6

Since HfTa_4C_5 and HECs have exceptional mechanical properties, high hardness, and super-high thermal stability, we investigate the thermal evolution of the plasmonic properties. Results for HfTa_4C_5 and the testbed system 3-HfNbTaTiZrC_5 are summarized in Figure 5. Panel (a) shows the experimental and theoretical EELS of spectra HfTa_4C_5 at different temperatures, in the range $T \in [300\text{--}1500]$ K. Increasing temperature produces minor changes to the main plasmonic properties: the spectral feature corresponding to the low-energy plasmon remains clearly recognizable up to $T=1500\text{K}$. The plasmon resonance is surprisingly stable even at high temperature, much higher than the standard plasmonic metals melting points (e.g., Ag and Au). Besides, the increased temperature causes an expected slight reduction of the intensity and a small broadening of the E_0 peak, which may be attributed to an increase of interband effects. Simulations (inset, panel a) concur with experiments, representing plasmonic resonance even at high temperatures. It is worth noting that here the temperature is the conformational temperature used to average the POCC ensemble [41]. Beyond 1500K , it might be arduous to characterize EELS due to equipment

limitation. Experimental-computational agreement for HfTa_4C_5 makes us confident of the existence of plasmonic properties of 3-HfNbTaTiZrC_5 even at ultra-high temperature (panel (b)) — temperature should have a minor effect on the spectral feature, causing a small reduction of the maximum intensity of E_0 , along with a broadening of the peak due to larger scattering effects.

Page7 – all new figure

Figure 5. Thermal evolution of plasmonic properties: (a) Experimental EELS spectra of HfTa_4C_5 as a function of temperature. Inset reports the corresponding theoretical spectra, evaluated at the same temperatures. (b) Simulated EELS spectra of 3-HfNbTaTiZrC_5 at different temperatures.

Page8 – Method section

Experiments: sample preparation. Tantalum carbide (99.5 % purity, Stanford Advanced Materials) and hafnium carbide (99.0% purity, H.C. Starck) were blended at a 4:1 ratio, respectively, in a Nalgene plastic jar with 3/16” WC-Co satellites and ball-milled at a 1:1 ball-to-powder ratio for 24 hours. Then, bulk HfTa_4C_5 was sintered using field assisted sintering technology (FAST), which allows for sintering at high heating rates and short processing times without sintering aids [73]. A 25 Ton FAST system (FCT Systeme GmbH) at the Penn State Applied Research Laboratory was used for sintering the powders. The powders were sintered in a 40 mm OD graphite die to a final pellet thickness of approximately 4 mm. The sintering was carried out in two concurrent steps at temperatures (2100°C/2400°C), pressures (55 MPa/40 MPa), and hold times (40 min/30 min) and was completed at a uniform heating rate of 100°C/min and under vacuum at ~3 mTorr. Density was measured to be 94.4% of the theoretical value using the Archimedes principle on a precision digital analytical balance (AND HM-202, ±0.1 mg).

Experiments: EELS spectra. Cross-sectional scanning transmission electron microscopy (STEM) and EELS were performed using an aberration corrected ThermoFisher Titan³ G2 60-300 with a monochromator and an X-field emission gun source at a beam energy of 300 keV. The HfTa_4C_5 specimen used for experimental EELS data acquisition was prepared according to Figure S9 of the Supplementary Information. Spectral resolutions of ≤0.2 eV were achieved for all EELS measurements as calculated by the full width at half maximum (FWHM) of the zero loss peak

(ZLP). Low loss EELS spectra were collected from HfTa_4C_5 as a function of temperature, ranging from room temperature ($\sim 25^\circ\text{C}$) to 1200°C , where the specimen was heated at a uniform rate of $10^\circ\text{C}/\text{second}$. EELS spectra were collected after stabilizing at each target temperature. A power-law decay function was fitted to the tail of the ZLP in front of the first absorption feature in order to filter out ZLP background signal and resolve the features of interest in the spectra. This was carried out using the Gatan DigitalMicrograph software suite.

Supplementary Information

EXPERIMENTAL MEASUREMENTS

Electron transparent lamellae were prepared for EELS analysis from the center of a cross-sectioned HfTa_4C_5 pellet in a Thermofisher Scientific Helios NanoLab 660 Dual Beam focused ion beam (FIB)-scanning electron microscope (SEM) using a 30 keV Ga ion beam. A sacrificial carbon protective layer was deposited on top of the HfTa_4C_5 prior to milling to limit excess ion implantation as the cross section was thinned to electron transparency, after which each face of the lamellae was polished using a 5 keV Ga ion beam to remove resputtered material and reduce the thickness of any amorphous implantation layer produced during milling at high accelerating voltages. Once thinned, the sections were lifted and transported to a Protochip Fusion Select Heating E-chip and micro-welded to the chip using localized 5 keV ion assisted deposition of carbon. Figure S9 shows the thinned and polished cross section after transport to the heating chip. Panel b) shows a monochromated STEM image of the HfTa_4C_5 lamellae, which indicates the region where all EELS spectra were collected as a function of temperature.

Figure S9. a) Ion milled electron transparent cross-section of sintered HfTa_4C_5 pellet fixed to Protochip Fusion Select Heating E-chip. The center of the cross-section was thinned to electron transparency and polished using a Ga ion beam prior to transfer to the E-chip, where the section was micro-welded to the chip using Ga ion assisted deposition of carbon as indicated on the left of the figure. b) Monochromated STEM micrograph of the HfTa_4C_5 cross-section, which indicates the region where all EELS spectra were collected.

Reviewer #3 (Comments to the Authors)

Q3. The authors have addressed the review comments satisfactorily, therefore, I recommend Accept.

Authors:

A3. We thank the reviewer for their assessment of our work.

END OF REPORT